# microRNA as a Maternal Marker for Prenatal Stress-Associated ASD, Evidence from a Murine Model

**DOI:** 10.3390/jpm13091412

**Published:** 2023-09-20

**Authors:** Taeseon Woo, Candice King, Nick I. Ahmed, Madison Cordes, Saatvika Nistala, Matthew J. Will, Clark Bloomer, Nataliya Kibiryeva, Rocio M. Rivera, Zohreh Talebizadeh, David Q. Beversdorf

**Affiliations:** 1Interdisciplinary Neuroscience Program, University of Missouri, Columbia, MO 65211, USA; tw2954@nyu.edu; 2Department of Biological Science, University of Missouri, Columbia, MO 65211, USA; ckkxd@mail.missouri.edu (C.K.); mcordes2642@gmail.com (M.C.); 3Department of Psychological Sciences, University of Missouri, Columbia, MO 65211, USA; niac86@umsystem.edu (N.I.A.); willm@missouri.edu (M.J.W.); 4Rock Bridge High School, Columbia, MO 65203, USA; nistala.saatvika@gmail.com; 5Genomics Core, University of Kansas Medical Center, Kansas City, KS 66160, USA; 6College of Bioscience, Kansas City University, Kansas City, MO 64106, USA; nkibiryeva@kansascity.edu; 7Division of Animal Sciences, University of Missouri, Columbia, MO 65211, USA; riverarm@missouri.edu; 8American College of Medical Genetics and Genomics, Bethesda, MD 20814, USA; ztalebizadeh@acmg.net; 9Department of Radiology, Neurology, and Psychological Science, William and Nancy Thompson Endowed Chair in Radiology, University of Missouri, Columbia, MO 65211, USA

**Keywords:** serotonin, microRNA, stress, autism spectrum disorder, SERT

## Abstract

Autism Spectrum Disorder (ASD) has been associated with a complex interplay between genetic and environmental factors. Prenatal stress exposure has been identified as a possible risk factor, although most stress-exposed pregnancies do not result in ASD. The serotonin transporter (SERT) gene has been linked to stress reactivity, and the presence of the SERT short (S)-allele has been shown to mediate the association between maternal stress exposure and ASD. In a mouse model, we investigated the effects of prenatal stress exposure and maternal SERT genotype on offspring behavior and explored its association with maternal microRNA (miRNA) expression during pregnancy. Pregnant female mice were divided into four groups based on genotype (wildtype or SERT heterozygous knockout (Sert-het)) and the presence or absence of chronic variable stress (CVS) during pregnancy. Offspring behavior was assessed at 60 days old (PD60) using the three-chamber test, open field test, elevated plus-maze test, and marble-burying test. We found that the social preference index (SPI) of SERT-het/stress offspring was significantly lower than that of wildtype control offspring, indicating a reduced preference for social interaction on social approach, specifically for males. SERT-het/stress offspring also showed significantly more frequent grooming behavior compared to wildtype controls, specifically for males, suggesting elevated repetitive behavior. We profiled miRNA expression in maternal blood samples collected at embryonic day 21 (E21) and identified three miRNAs (mmu-miR-7684-3p, mmu-miR-5622-3p, mmu-miR-6900-3p) that were differentially expressed in the SERT-het/stress group compared to all other groups. These findings suggest that maternal SERT genotype and prenatal stress exposure interact to influence offspring behavior, and that maternal miRNA expression late in pregnancy may serve as a potential marker of a particular subtype of ASD pathogenesis.

## 1. Introduction

Autism Spectrum Disorder (ASD) is a complex and heterogeneous neurodevelopmental disorder that affects an estimated 1 in 44 children in the United States [1]. It is characterized by deficits in social communication and interaction, as well as restricted and repetitive patterns of behavior, interests, or activities [2]. The etiology of ASD is not yet fully understood, but evidence suggests that a combination of genetic and environmental factors may contribute to its development [3,4]. Prenatal stress has been identified as a possible environmental risk factor for ASD, although most stress-exposed pregnancies do not result in ASD [5,6,7]. The association between prenatal stress exposure and neurodevelopmental disorders, including ASD, has been implicated in alterations in gene expression involved in the regulation of the hypothalamic–pituitary–adrenal (HPA) axis, which regulates the body’s response to stress [8,9].

One gene that has been linked to prenatal stress and HPA axis dysregulation is the serotonin transporter (SERT) gene, which plays a key role in the regulation of serotonin neurotransmission [10,11]. The SERT gene has a functional polymorphism known as the serotonin transporter-linked polymorphic region (5-HTTLPR), which has been linked to stress sensitivity and cortisol elevation associated with stress reactivity in individuals carrying the short (S) allele and displaying low SERT gene methylation [12,13]. The presence of the S allele of 5-HTTLPR has been shown to mediate the association between maternal stress exposure and ASD in humans [5,14]. Further understanding of the mechanisms underlying the association between SERT and stress-related disorders may shed light on the pathogenesis of one subtype of ASD and inform the development of novel preventative or therapeutic strategies.

Recent studies have suggested that alterations in microRNAs (miRNAs) may also play a role in the association between prenatal stress exposure and neurodevelopmental disorders [15,16,17,18]. miRNAs are small non-coding RNA molecules that regulate gene expression by targeting specific mRNA transcripts for degradation or translational repression [19]. A growing body of evidence suggests that miRNAs are involved in the regulation of neural development and that alterations in their expression may contribute to the pathophysiology of neurodevelopmental disorders [20,21,22]. Several studies have investigated the role of miRNAs in the regulation of neurodevelopmental processes and have identified potential candidate miRNAs that may be important in ASD [23]. Differential expression of several miRNAs was observed in post-mortem brain tissue samples of individuals with ASD compared to control samples [24]. Additionally, in the frontal cortex, Brodmann area 10, miR-142-5p, miR-142-3p, miR-451a, miR-21-5p, and miR-144-3p showed higher expression in small RNA sequencing of ASD post-mortem brain tissue [23,25]. Studies combining cognitive activity and neuroimaging have demonstrated that the prefrontal cortex plays a key role in the regulation of social cognition, particularly in the analysis of emotions [26].

Our previous study has also investigated the differential expression of maternal blood miRNA as a potential indicator of gene–environment interactions involved in the development of ASD, particularly in the context of prenatal stress exposure and SERT gene polymorphisms [27]. In that study, we investigated the effects of prenatal stress and the SERT gene on maternal blood miRNA expression among mothers with children with ASD, providing evidence for epigenetic alterations in relation to a gene–environment interaction model in ASD. In our gene–environment interaction animal model, we exposed SERT heterozygous knockout (SERT-het) mice to chronic variable stress (CVS) during pregnancy, which has previously been demonstrated to result in altered social interest and social preference in the mouse model [28]. Furthermore, we profiled epigenetic markers, including miRNAs, whose alterations coincided with neurodevelopmental changes associated with the gene–environment interaction in ASD. We identified miRNAs that were upregulated in the brains of offspring in mice, some of which overlapped with the miRNAs identified in the blood of mothers of children with ASD in our clinical samples [27,29]. It is crucial to investigate the miRNA profile during pregnancy in maternal blood due to the vital function of blood in transmitting maternal stress to the fetus through the placenta and the ability of miRNAs to traverse the placental barrier [30]. A unique miRNA profile may serve as a potential biomarker for a particular subtype of ASD pathogenesis.

Given the potential role of miRNAs in neurodevelopmental processes, we used the SERT-het/stress mouse model, which was established based on the findings from our previous study [28], to investigate the effects of maternal stress exposure and SERT genotype on miRNA expression in maternal blood during pregnancy. The SERT-het mouse model used in this study recapitulates the human population of S-allele carriers, making it a relevant model for studying the gene–environment interaction in ASD [31]. We hypothesized that maternal stress exposure and SERT genotype would interact to alter miRNA expression in maternal blood and that these changes in miRNA expression may eventually serve as a potential biomarker of ASD risk during pregnancies associated with stress.

In a mouse model, we investigated the effects of prenatal stress exposure and maternal SERT genotype on offspring behavior and explored the potential of maternal miRNA expression as a potential marker of ASD risk during pregnancy, focusing on the effect of maternal genotype, as with our previous work [27,28,29]. Time points E21 and PD60 were chosen as they coincide with critical periods of neurodevelopment in mice and are also equivalent to the third trimester of human pregnancy and early childhood, respectively [32]. Importantly, postnatal day 60 in mice is a critical time point for assessing ASD-associated behavior, as our previous study in humans identified differentially expressed maternal blood miRNA profiles at the age of 7 years old in ASD children, allowing us to examine how the miRNA profile in the human maternal sample might correspond to the profile observed in the mouse maternal sample after a corresponding timeframe of offspring developments [27].

## 2. Materials and Methods

### 2.1. Animals

Male homozygous serotonin transporter (SERT) knockout (KO) mice with a C57BL6/J background and male/female C57BL6/J mice were purchased from Jackson Laboratories (Bar Harbor, ME, USA). The breeding protocol used in this study involved mating one homozygous SERT-KO male mouse with two wildtype (WT) female mice to generate the SERT-het female dams as described previously [28,33]. Eight-week-old experimental dams were paired with WT males and inspected for a vaginal plug the following morning. Identification of a plug was designated as gestational day 0. All pups were weaned on postnatal day 21 (PD21), with behavior testing beginning on PD60, and all offspring were ultimately sacrificed around PD 70. Animals were maintained in a temperature- and humidity-controlled room at 25 ± 2 °C on a 12-h light/dark cycle with food and water available *ad libitum*. All animals were housed in clear carbonate cages provided with aspen shaving bedding. All procedures were in accord with protocols approved by the University of Missouri Institutional Animal Care and Use Committee (IACUC #20960). 

### 2.2. Prenatal Chronic Variable Stress 

SERT-het and wildtype female mice were categorized into four groups based on the presence or absence of chronic stress exposure during pregnancy, including wildtype no stress (WN), wildtype stress (WS), SERT-het no stress (HN), and SERT-het stress (HS) groups. Stress exposure for SERT-het mice started at gestational day 6 and continued until parturition for all mice in the stress groups. Stressors used in this study included constant light exposure (36 h), overnight novel noise exposure (radio static), exposure to novel objects (marbles), fox urine exposure (1 h), multiple cage changes in one day (3 times throughout the light cycle), and restraint for a duration of 10 min. The stress protocol of six stressors was repeated approximately 2.5 times or until parturition, consistent with our previous work [28,33]. The selected stressors were chosen specifically for their ability to induce stress without causing pain, affecting food intake or weight gain, as previously described [34].

### 2.3. Blood Collection, Total RNA Isolation, and miRNA Expression Profiling

On embryonic day 21 (E21), blood samples were collected from pregnant mice under isoflurane anesthesia by drawing approximately 500 μL of blood from the heart using a heparinized syringe. The collected blood was then immediately inverted 10 times in a 2-mL tube containing 1.3 mL of RNAlater solution from Ambion (Grand Island, NY, USA) and kept on ice. The RNAlater blood samples were stored at −80 °C until analysis. At postnatal day 60 (PD60), the dams of the behaviorally tested offspring were sacrificed in a similar fashion in order to collect and compare the miRNA profiles of dams at E21 versus PD60. The Genomics Core at the University of Kansas Medical Center’s Microarray Facility began with total RNA that was isolated from 1 mL of the RNAlater blood samples using the Mouse RiboPure-Blood RNA Isolation kit from Ambion. Quality control of the total RNA isolates was performed using the TapeStation RNA ScreenTape assay (Agilent 5067-5576, Santa Clara, CA, USA). Biotin labeling of Total RNA (1 μg) was performed using the FlashTag Biotin HSR RNA Labeling Kit (Life Technologies 901910, Thermo Fisher Scientific, Waltham, MA, USA). The FlashTag labeling system employs the 3DNA dendrimer signal amplification technology. 3DNA dendrimer is a branched structure of single- and double-stranded DNA conjugates that incorporates numerous biotin labels for ultrasensitive expression detection. The Genomics Core used the GeneChip system for processing the miRNA expression arrays. Using the GeneChip 645 Hybridization Oven, labeled target RNA is hybridized overnight (16 h, 48 C, 60 rpm) to interrogate oligo probes contained in the GeneChip miRNA 4.0 expression cartridge array (Life Technologies #902412). Hybridized GeneChip arrays undergo low and high stringency washing and R-Phycoerythrin-Streptavidin staining procedures using the GeneChip Fluidics Station 450 running the FS450_0002 Fluidics profile. After washing, GeneChip arrays are processed using a single scan on the GeneChip Scanner 3000 7 G with an autoloader. Fluidics and scan functions are controlled by Affymetrix GeneChip Command Console software (AGCC). Raw expression data are loaded onto the Microarray Data Management System (MDMS) for access by the Investigator and further data analysis.

We used Affymetrix^®^ Transcriptome Analysis Console (TAC) software (Version 3.1.0.5, Thermo Fisher Scientific, Waltham, MA, USA) to identify differentially expressed (DE) miRNAs by setting *p*-value and fold change thresholds (one-way analysis of variance: *p* < 0.05 and absolute fold change ³1.2). The expression profiles across samples were compared and visualized using a Venn diagram to highlight similarities and differences. 

### 2.4. Behavioral Assays

At PD60, as with previous work [35], we assessed the behavior of the offspring using a battery of tests to evaluate sociability, repetitive and anxiety-like behaviors. All behavioral tests were conducted during the light phase of the cycle with minimal lighting, and each apparatus was cleaned with isopropyl alcohol between subjects. Tests were recorded using video-tracking software, Stoelting AnyMaze Software (Wood Dale, IL, USA).

### 2.5. Social Approach

Social preference was evaluated using the three-chamber test, which consists of a Plexiglas apparatus (54.5 cm × 41.5 cm × 22 cm) with three chambers (17.5 cm × 41.5 cm × 22 cm) separated by dividers (0.5 cm thick walls) with openings (10.25 cm × 2.5 cm) for free movement between chambers [28,33,36]. Experimental mice were first familiarized with the apparatus for 10 min. A stranger mouse was then placed in a wire cage on one side of the apparatus to assess social approach, while the opposite chamber remained empty. Sides were counterbalanced with each trial as each mouse either had the familiar placed in the right or left chamber. The experimental mouse was allowed to explore the three chambers for an additional 10 min. Time spent interacting with this single stranger option is used as an index of sociability. Lastly, a new stranger mouse was placed in a wire cage in the opposing chamber while the original stranger mouse remained in its first chamber, and time spent with the novel stranger is indicative of social behavior associated with social novelty and a willingness to explore new social interactions. The experimental mouse was then placed in the three-chamber apparatus for another 10 min. The video tracking software measured the duration of time spent by the testing mice in each chamber, as well as their proximity to both the novel object and stranger mice. 

### 2.6. Open Field

The open field test was conducted the following day after the 3-chamber test to evaluate general locomotor and exploratory behavior. The mice were placed in a Plexiglas enclosure measuring 45 cm × 45 cm × 22 cm, and each mouse was tested individually for 20 min. AnyMaze software was used to record the total distance traveled by each mouse and the time they spent moving. Additionally, time spent within the inner region of the open field apparatus was measured to quantify thigmotaxis, a validated behavioral measure of anxiety in mice that prefer to explore peripherally when placed in this type of apparatus [37]. The amount of time spent following the walls of the apparatus may be interpreted as anxiety and is measured by the time not spent within the inner region. 

### 2.7. Elevated-plus Maze

The elevated-plus maze measures anxiety-like behavior to determine if there are any effects due to general anxiety [38]. The maze is raised off the ground (75 cm) and has two open arms (35 cm × 6.25 cm × 0.25 cm) and two enclosed arms (35 cm × 6.25 cm × 21 cm) which join at an intersection (5 cm × 5 cm). At the start of the experiment, mice were placed in the intersection of the maze, and their activity was recorded for 10 min. The amount of time the mouse spent in the open arms and closed arms was recorded and used to calculate the open-arm ratio. The less time spent on the open arms and, thus, a lower open arm ratio indicates increased anxiety observed in the mouse being tested. 

### 2.8. Repetitive Behavior

Repetitive behavior was assessed using two different approaches. First, during the open field test, self-grooming behavior was measured, as excessive grooming has been linked to repetitive behavior in animal models of neuropsychiatric disorders, including ASD [39]. To assess self-grooming behavior, we counted the total time spent grooming and the frequency of grooming events during the 20-min testing period. These measures have been used in previous studies to assess repetitive behavior in mice [33].

Second, marble burying was used to assess repetitive behaviors [40,41]. The test was conducted in a cage with corncob bedding covering the bottom. Mice were habituated in the cage for 20 min, after which 20 glass marbles were placed in a 5 × 4 arrangement on top of the bedding. The animals were allowed to explore the marbles for 20 min, after which the number of marbles buried under the bedding was recorded. Marbles were considered buried if more than two-thirds of the marble was covered by bedding.

### 2.9. Statistics

Data were analyzed using R Studio Version 2022.12.0+353 [42]. Two-way ANOVA (maternal genotype x stress exposure) was conducted for outcome variables, but two-tailed Student’s *t*-tests were used to analyze a priori hypotheses. Other statistical comparisons were performed using Tukey HSD post-hoc testing. Results were considered significant at *p* < 0.05. All written results include descriptive statistics reported as mean (*M*) and standard error of mean (*S.E.M.*). However, included figures are depicted using box plots and include a given group’s median, first, and third quartiles, as well as whiskers that extend to a maximum of 1.5 times the interquartile range (where the interquartile range is the distance from the first to third quartiles). Due to issues during behavioral testing and data collection, some outcome measures may have reduced sample sizes in comparison to others, and will be noted when discussed in the results section.

## 3. Results

Total offspring assessed included male (*n* = 83; HS *n* = 27; WN *n* = 23; WS = 14; HN *n* = 19) and female (*n* = 42; HS *n* = 10; WN *n* = 7; WS *n* = 11; HN *n* = 14) mice. The authors note this is an abnormal sex ratio (proportion of male-to-female births). While we expected closer to 1:1, several dams birthed highly male-biased litters, driving our proportions. The sex ratio of rodents can be affected by a multiplicity of factors, including maternal diet, hierarchy placement of the mother, and hormone exposure, amongst other factors [43,44]. One possible contributor to having an increased proportion of males may be due to our breeding during the warmer seasons, where one study showed mice produced more males per litter during the spring and summer, which was attributed to the higher availability of food for the litter that would typically be associated with these seasons [45]. 

### 3.1. Social Approach

#### 3.1.1. Novel Stranger versus Empty Chamber

The three-chamber social approach testing included 83 male mice and 42 female mice. To measure sociability, time spent within proximity of the novel stranger was contrasted with time spent investigating the empty chamber. These comparisons are presented in the form of a ratio and are referred to as the social preference index (SPI). When comparing SPI across groups, we see a significant main effect of genotype (F_1,79_ = 13.95, *p* < 0.001, *η*^2^ = 0.14, see Figure 1C) amongst males. Aligning with a priori hypotheses, HS born males (*M* = 0.517, *S.E.M.* = 0.017) showed a significantly (*t*_46.103_ = −4.8882, *p* < 0.001, unpaired *t*-test) lower SPI when compared to WN born males (*M* = 0.639, *S.E.M* = 0.019). Follow-up two-way ANOVA depicted a significant main effect of genotype amongst the male groups (F_1,79_ = 6.85, *p=* 0.01, *η*^2^ =0.08, see Figure 1A), and female groups (F_1,38_ = 5.66, *p <* 0.05, *η*^2^ = 0.13, see Figure 1B) in time spent within proximity of the novel stranger, males (*n* = 27) born to HS dams exhibited (*M* = 157.10, *S.E.M.* = 7.34) significantly less (*t*_37.096_ = −3.7404, *p* < 0.001, unpaired *t*-test) time interacting with the novel stranger compared to males (*n* = 23) born to WN dams (*M* = 209.82, *S.E.M.* = 12.03). Since we must consider the time spent in the middle chamber for the following measure, we assessed the time spent in the middle chamber here as well; our analysis revealed a main effect of stress in the time spent in the center chamber in male mice (F_1,46_ = 11.35, *p* < 0.001, *η*^2^ = 0.19). The reduction in time in females (*n* = 10) born to HS dams (*M* = 158.3, *S.E.M.* = 14.79) when compared to females (*n* = 7) born to WN dams (*M* = 191.01, *S.E.M.* = 12.10), did not reach significance, (Figure 1B). Moreover, males born to HS dams exhibited a lower preference for social interaction with the novel mice than males born to WN dams in the manner of total time within the social approach zone and SPI (Figure 1C). In contrast, female mice born to HS dams did not display lower sociability behavior when compared to those born to WN dams. 

#### 3.1.2. Novel Stranger versus Familiar Stranger

The social approach task was then followed by a social novelty task in the 3-chamber apparatus to assess the amount of time a given mouse spent interacting with the same familiar mouse from the previous social approach task or interacting with a novel stranger. Those from control conditions have been routinely shown to spend significantly more time investigating the novel stranger as opposed to spending their time with the previous, familiar mouse [41,46]. Like previous indices, analyses of this trial included the ratio of time spent with the novel mouse compared to the total time spent with the familiar and novel mice, denoted as the social novelty index (SNI). While males did not show any significant main effects in SNI, females revealed a main effect of genotype (F_1,38_ = 4.45, *p* < 0.05, *η*^2^ = 0.10, see Figure 2C) in SNI scores. While those of SERT lineage had higher SNI scores (HS: *M=* 0.63 ± 0.04; HN: *M* = 0.64 ± 0.04), this seems to be primarily driven by these female offspring spending less time on average in the familiar stranger zone (HS: *M* = 90.77 ± 12.29; HN: *M* = 84.51 ± 8.17) compared to the wildtypes (WS: *M* = 131.52 ± 15.14; WN: *M* = 144.04 ± 19.33). Furthermore, no significant main effects in genotype nor stress were seen in time spent interacting with the novel stranger in male nor female groups; (see the right panels of Figure 2A,B). As is characteristic of autism mouse models, decreased social behaviors may suggest the avoidance of both the familiar and novel mice in each respective chamber. Thus, researchers also examined the time spent in the center chamber relative to time spent in the chambers of the familiar and novel mice. While the experimental mice still spent most of the testing period in the possible side chambers, there was a significant main effect of stress (F_1,46_ = 6.64, *p* < 0.05, *η*^2^ = 0.12) when examining the time spent within the center chamber amongst the male offspring, which seems to be primarily driven by the difference between HS offspring (*M* = 134.65 ± 7.93) and HN offspring (*M* = 89.19 ± 9.39). However, this increased use of the center chamber is not restricted to just the social novelty task, as it also occurred during the social approach assay, so we cannot assume that increased time in the central zone is only due to avoidance of social behaviors as the social approach trial only had one chamber of the three occupied by a social stimulus (novel mouse).

### 3.2. Elevated-plus Maze

Open arm (OA) ratio was used to quantify anxiety-like behavior in male (*n* = 67) and female mice (*n* = 42). Primary two-way ANOVAs did not show any significant main effects in the OA ratio, time spent on the open arm of the apparatus, or time spent in the closed arm of the apparatus for male subjects. Females, on the other hand, did show a significant interaction effect between genotype and stress (F_1,38_ = 7.41, *p* < 0.01, *η*^2^ = 0.16, see Figure 3) in their respective OA ratios. Post-hoc comparisons indicate this was primarily driven by less time spent in the open arm with stress as compared to non-stressed in the wildtype female mice (*p* < 0.05). Female offspring also showed a main effect of genotype (F_1,38_ = 4.26, *p <* 0.05, *η*^2^ = 0.09) in time spent in the closed arm and a significant interaction effect between genotype and stress (F_1,38_ = 6.20, *p* < 0.05, *η*^2^ = 0.14) in time spent in the open arm of the apparatus. Female HS (*n* = 10) offspring spent less time in closed arms (*M* = 332.80, *S.E.M* = 23.50) and more in open arms (*M* = 223.43, *S.E.M* = 20.92) than did the female WS (*n* = 11) offspring (*M* = 399.21, *S.E.M* = 24.32; *M* = 172.78, *S.E.M* = 24.45) on average, but neither reached significance (*p* = 0.06; *p* = 0.13).

### 3.3. Open Field Tests

No significant main effects were seen amongst the males nor females tested in distance traveled and time immobile; (see Figure 4A,C). However, a two-way ANOVA revealed a significant main effect of prenatal stress amongst males (F_1,54_ = 8.247, *p* < 0.01, *η*^2^ = 0.13, see Figure 4B) in time spent within the inner region of the apparatus. Subsequent post-hoc testing revealed that males (*n* = 16) born to prenatally stressed SERT-het dams spent significantly more time (*M* = 208.67, *S.E.M.* = 18.07) in the inner region than did males (*n* = 19) born to control SERT-het dams (*M* = 155.39, *S.E.M.* = 7.37), suggesting less anxiety-like behavior (*t*_20_ = 2.73, *p* < 0.05, unpaired *t*-test) in prenatally stressed SERT-het male offspring. 

### 3.4. Marble Burying

In a two-way ANOVA, males exhibited a significant main effect of stress (F_1,79_ = 13.66, *p* < 0.001, *η*^2^ = 0.15, see Figure 5A) during the marble burying assay. HS males (*n* = 27) displayed lower rates (*t*_33.682_ = 1.9394, *p* = 0.06, unpaired *t*-test) of marble-burying behavior (*M* = 6.04, *S.E.M* = 0.88) than HN males (*n* = 19) exhibited (*M* = 9.11, *S.E.M* = 1.26), which approached significance. WS (*n* = 14) males also showed lower (*t*_34.95_ = 3.8143, *p <* 0.001, unpaired *t*-test) marble-burying behavior (*M* = 3.36, *S.E.M.* = 0.80). than did their WN (*n* = 23) counterparts (*M* = 8.26, *S.E.M.* = 1.00). Females did not show any significant main effects in marble-burying behavior across all groups. These results suggest lower rates of marble burying for the stressed groups compared to the non-stressed group, counter to what literature has reported previously [47]. Some have suggested that the decreased rates of marble burying may be due to a relative preference for self-grooming behaviors, as observed in other marble-burying ASD mouse models [48].

### 3.5. Spontaneous Self-Grooming

Males exhibited a significant main effect of stress (F_1,75_ = 7.82, *p* < 0.01, *η*^2^ = 0.09, see Figure 5B) on grooming behavior frequency in a two-way ANOVA. HS males (*M* = 5.44, *S.E.M* = 0.46) exhibited increased grooming behavior (*t*_38.87_ = 3.1803, *p* < 0.01, unpaired *t*-test) when compared to WN males (*M* = 3.79, *S.E.M* = 0.25, *n* = 19). However, there were no significant main effects amongst the male groups when looking at total grooming duration, see Figure 5C. Female offspring tested did not show any significant main effects in grooming frequency and, subsequently, no significant differences in grooming duration.

### 3.6. Possible miRNA Biomarkers

Our primary differential expression analysis from maternal blood collected on E21 revealed significant variations in miRNA expression across our four distinct conditions (HS, HN, WS, WN). Out of 3195 miRNAs detected, we identified 51 differentially expressed miRNAs between HS and WN, 61 between HS and WS, and 72 between HS and HN. Three miRNAs—*mmu-miR-5622-3p*, *mmu-miR-6900-3p*, and *mmu-miR-7684-3p*—exhibited consistent upregulation in the HS condition across all comparisons (see Table 1). These data suggest that stress conditions in SERT-het mice may influence miRNA expression profiles, with potential implications for understanding the molecular mechanisms of the stress response.

The second dam blood collection for miRNA isolation was conducted on PD60 after all behavioral testing in the offspring had been completed. Amongst the same 3195 analyzed, we identified 103 differentially expressed miRNAs between HS and WN, 169 between HS and WS, and 132 between HS and HN. When collapsing across these lists, only six miRNAs—*mmu-miR-16-5p*, *mmu-miR-1893*, *mmu-miR-6347*, *mmu-miR-126a-3p*, *mmu-miR-340-5p*, and *mmu-miR-3620-3p*—was observed in all three groups (see Table 1); the first three listed were downregulated while the latter three were upregulated. Thus, markedly different miRNA signatures than found in E21 samples, speaking to the dynamic nature of miRNA across time. 

To further contextualize the 3 miRNAs identified at E21, which is the focus of this study, aiming to understand the epigenetic environment during pregnancy, we collated lists of target prediction proteins from the miRDB online database of miRNA targets. The predicted targets of *mmu-miR-7684-3p*, *mmu-miR-5622-3p*, and *mmu-miR-6900-3p* are numerous, with approximately 921 unique target genes identified by the miRDB prediction database [49]. The top-ranked predicted target genes for each miRNA are Zfp148, Fgfbp3, and Shisa7, respectively, while they have not been experimentally validated yet [50]. We compared our list of gene targets to the SFARI Gene database, which collects information on genes linked to ASD, and found 96 that overlapped (about 10% of predicted interactions). We also reviewed gene projections from both decreased and increased miRNAs at 60 days after birth. In these groups, around 10% (201 and 132 genes, respectively) were also found in the SFARI Gene database [51]. 

Using the PANTHER Classification System [52,53] and shinyGO 0.77 analyses [54], we examined the functions of these projections based on Gene Ontology (GO) biological processes. At embryonic day 21, PANTHER identified that most genes that were targeted were in specific subclasses of biological processes: the glucocorticoid receptor signaling pathway (GO:0042921), regulation of the B cell receptor signaling pathway (GO:0050855), and the positive control of bone formation (GO:0045778). With shinyGO, we identified that the most enriched categories are related to CNS development (specifically the brain, head, and forebrain—see Figure 6; Appendix A), but a large proportion of genes were related to the positive regulation of biosynthetic and metabolic processes.

### 3.7. Behavioral Correlations

In our study, miRNA expression did not reveal a significant association with social behaviors across males and females. However, there was a trend towards an association between grooming duration in female mice and miRNAs 5622-3p and 7684-3p (r = 0.90, *p* = 0.09; r = 0.91, *p* = 0.08, respectively). While none of the above correlation coefficients were found to be statistically significant, there are definitive overall trends that formed when contrasting fold change to behavioral measures. Also, considering that we used group averages to build these correlations, these results cannot account for large shifts in variability on the individual level but rather provide an estimated relationship that should be further researched.

## 4. Discussion

In this study, we found that the social preference time of SERT-het/stress offspring was significantly lower than that of wildtype control offspring, indicating reduced sociability in males. Female offspring of SERT background had higher social novelty index scores than their WT counterparts, but male offspring groups had no such differences. Additionally, SERT-het/stress male offspring showed significantly increased grooming frequency compared to wildtype control offspring, suggesting elevated repetitive behavior. SERT-het/stress females showed a similar increase compared to controls but did not reach significance. Therefore, the model affected social behavior in both sexes but had a preferential effect on repetitive behavior in males. Though our marble burying results were opposite of expectations (an increase in repetitive behavior is often shown in increased number of marbles buried), the preference for repetitive behavior could impact this behavior, and future studies should monitor the different repetitive behaviors that can be expressed during this task [47,48,55,56,57,58,59,60]. Our findings are consistent with previous research demonstrating that maternal stress during pregnancy can increase the risk of ASD and other neurodevelopmental disorders in offspring, particularly when combined with genetic vulnerability [28,33]. Furthermore, our results identified three miRNAs *(mmu-miR-7684-3p*, *mmu-miR-5622-3p*, *mmu-miR-6900-3p*) that were differentially expressed at E21 in the SERT-het/stress group compared to all other groups. This suggests that alterations in miRNA expression in maternal blood may serve as a marker of ASD risk during pregnancy in the context of prenatal stress and genetic stress susceptibility, offering a potential avenue for early detection and intervention. The miRNA changes were not found to correlate with the behavioral changes, but only group analyses could be performed for this comparison, as the behavior could not be assayed in the same mice with which blood was taken from the dam at E21, limiting the possibility of the greater statistical power of within-subject comparison.

miRNAs have been identified as potential biomarkers for pregnancy-related complications, and their dysregulation in maternal blood has been suggested as a means of identifying risk factors for such complications [30,61,62,63]. While miRNAs play key roles in placental development and function, placental-specific miRNAs can also be detected in maternal plasma and have been found to be dysregulated in women with pregnancy-associated complications [64]. However, blood miRNAs have the advantage of being easily accessible, making them a promising tool for early detection and monitoring of pregnancy complications [65]. Our previous study found that altered expression of miRNAs in maternal blood is associated with the interaction between prenatal stress and serotonin transporter (SERT) genotype in a clinical population, indicating the potential for epigenetic regulation in the development of ASD [27].

The three upregulated miRNAs at E21 exhibited only slight changes in expression levels, with fold changes of around 1.5 (see Table 1). However, even small changes in expression levels may have important functional implications, given the regulatory role that miRNAs play in gene expression [66,67].

Previous studies have suggested that altered miRNA expression may contribute to the development of neurodevelopmental disorders, including ASD [27,68], but the exact mechanisms by which miRNAs impact neurodevelopment are still not well understood. It is possible that the miRNAs identified in our study directly impact neurodevelopment in offspring, suggested by the increased enrichment of pathways related to CNS development, but it is also possible that the changes in miRNA expression are inherent to the mother’s gene expression, or perhaps are indirectly modulating unknown fetal programming mechanisms, rather than directly affecting the developing fetus. Another plausible hypothesis could be the miRNAs identified may have an effect on the manipulation of the placental bridge. Future studies should investigate the functional role of these miRNAs in the context of neurodevelopment and explore if there is any utility of these miRNAs as therapeutic targets.

While our study suggests a potential for miRNAs as markers for identifying ASD risk during pregnancy associated with prenatal stress and genetic stress susceptibility, further research is needed to determine the clinical relevance of these biomarkers in human populations in particular. Furthermore, such miRNAs need to be validated as potential ASD biomarkers in large and more diverse human populations, particularly those related to prenatal stress exposure. The dynamic nature of miRNA expression in blood during pregnancy has important implications for future research in human subjects [69]. In our study, we investigated miRNA expression in maternal blood samples collected during pregnancy. However, the miRNA expression in maternal blood samples when their offspring reached 60 days of age was observed to be considerably distinct from those detected during pregnancy, indicating miRNA expression changes throughout the course of gestation and well after birth. Our previous human studies of maternal blood samples were obtained when ASD children were around 7 years old [27], which is a compatible timeline with our mouse study at 60 days old [32]. Given changes in blood miRNAs across time, it is necessary to examine miRNA profiles throughout pregnancy to better understand their potential clinical utility as a biomarker for ASD risk in prenatal stress exposure. Future studies that investigate the potential of miRNA expression as a biomarker of ASD risk in human populations would need to assess a similar timeframe to determine whether our findings have clinical relevance.

In addition to investigating the potential of miRNA-based biomarkers, future studies should examine miRNA alterations in embryonic and 60-day-old offspring brains. Given that miRNAs can cross the placental barrier and impact fetal brain development, elucidating how specific maternal miRNAs are related to epigenetic alterations in the offspring’s brain is crucial for understanding the mechanisms underlying the association between prenatal stress exposure and ASD. Understanding the specific miRNA alterations at different developmental stages may provide insights into the potential mechanisms underlying the association between prenatal stress, SERT genotype, and ASD risk. It will also be important to investigate the specific miRNA expression changes in different regions of the brain, such as the striatum, and how these changes may contribute to the alterations in neural circuits and behavior observed in ASD, as our previous work has shown, that dopaminergic levels are altered in this region in the context of prenatal stress and SERT genotype [33]. Other epigenetic alterations in specific brain regions may provide valuable information regarding the contribution of gene–environment interactions to ASD risk. Therefore, future studies could investigate the specific epigenetic alterations associated with prenatal stress, SERT genotype, and the expression of differentially expressed miRNAs in specific brain regions in order to further elucidate the molecular mechanisms underlying ASD-like behaviors and potentially inform the development of novel therapeutic strategies.

Furthermore, additional investigation is needed to determine the specific epigenetic alterations in the placenta associated with prenatal stress and SERT genotype and how these changes may contribute to the development of ASD. The importance of the placenta, an endocrine tissue that facilitates fetal–maternal exchange, as a mediator of the effects of stress during pregnancy has become increasingly recognized [9,70,71]. Placental gene expression patterns have been shown to be altered by maternal stress exposure, and these changes may lead to long-term effects on the offspring’s neurodevelopment [9,72,73]. For instance, placental gene expression profiles have been linked to changes in brain morphology, altered stress reactivity, and behavioral outcomes in rodents and humans [74,75,76,77]. Recent studies have shown that placental miRNA expression patterns are altered by maternal stress exposure and can lead to long-term effects on offspring neurodevelopment [27,78,79]. Therefore, elucidating the specific molecular and cellular alterations in the placenta associated with prenatal stress and SERT genotype may provide additional insights into the pathogenesis of ASD and inform the development of novel preventative or therapeutic strategies. Thus, further studies are needed to determine the mechanisms by which miRNAs in maternal blood may contribute to the pathogenesis of ASD and whether these miRNAs are acting directly on the fetus as it crosses the placenta or indirectly by modulating placental gene expression. Additionally, subsequent research may be able to provide a greater understanding of the relationship between maternal miRNA samples with a more robust sample with designs allowing comparisons within the same animal.

Prenatal stress alone or SERT-het genotype alone had limited impact on offspring behavior; however, when these two factors were combined, a statistically significant alteration in offspring behavior was observed, indicating a potential gene–environment interaction in the development of ASD-like behaviors, as with our previous work [28,33]. The DE miRNAs observed in the wildtype/stress group may represent a protective mechanism in response to the hostile prenatal environment, potentially mitigating the negative effects of prenatal stress on offspring behavior. These miRNAs may be involved in regulating the expression of genes involved in the development of neural circuits or other critical processes disrupted by prenatal stress exposure.

Since all of our previous work has focused on the impact of maternal genotype [27,28,29,33], that was also the focus of the present study. Our sample was not sufficient to add another statistical comparison to examine the interaction of offspring genotypes, but this will be a critical future direction for understanding this effect.

Additionally, previous research has shown that docosahexaenoic acid (DHA) dietary supplementation during pregnancy and lactation can mitigate the negative effects of prenatal stress exposure on offspring behavior and epigenetic alterations in our gene–environment interaction mouse model [33]. Therefore, investigating the epigenetic alterations associated with DHA supplementation may provide additional insights into the potential mechanisms underlying the association between prenatal stress, SERT genotype, and ASD risk.

Overall, our study provides important insights into the potential role of miRNAs associated with ASD during pregnancy, particularly in the context of gene–environment interactions. Our findings suggest that alterations in miRNA expression in maternal blood during pregnancy are associated with autism-associated behaviors in the offspring, offering potential targets for intervention. Furthermore, our investigation of the SERT-het/stress mouse model sheds light on the complex interplay between genetic and environmental factors in the pathogenesis of ASD. Further research is necessary to confirm the utility of miRNA expression as a biomarker for ASD risk during pregnancy in this setting and to elucidate the mechanisms underlying the association between miRNAs, prenatal stress, and ASD. Ultimately, our study may contribute to the development of personalized diagnostic and therapeutic strategies for individuals with ASD and their families.

## Figures and Tables

**Figure 1 jpm-13-01412-f001:**
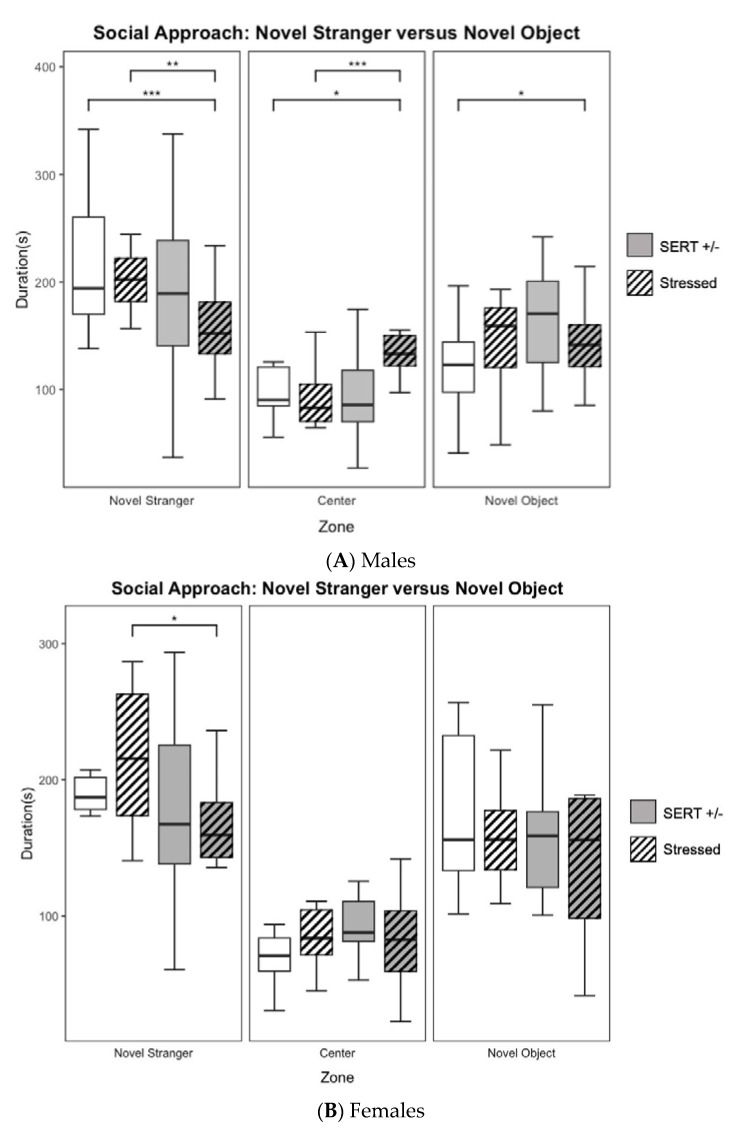
Social behavioral data obtained during the novel stranger versus novel object of the social approach task (**A**). A priori *t*-testing revealed that males born from HS dams spend less time interacting with the novel stranger than do wildtype controls as well as WS offspring. The HS males also spent significantly more time in the center chamber and interacting with the novel object than did the WN control. WS mice also spent significantly less time in the center chamber than the HS mice. (**B**). A priori *t*-testing showed HS females spent significantly less time interacting with the novel stranger than only the WN stressed mice. (**C**). Proportion of time spent with the novel stranger when compared to the novel object is quantified as the SPI in which male HS mice showed a lower preference for interacting with the novel stranger than did WN controls (* *p* < 0.05, ** *p* < 0.01, *** *p* < 0.001, **** *p* < 0.0001 for individual pairwise comparisons).

**Figure 2 jpm-13-01412-f002:**
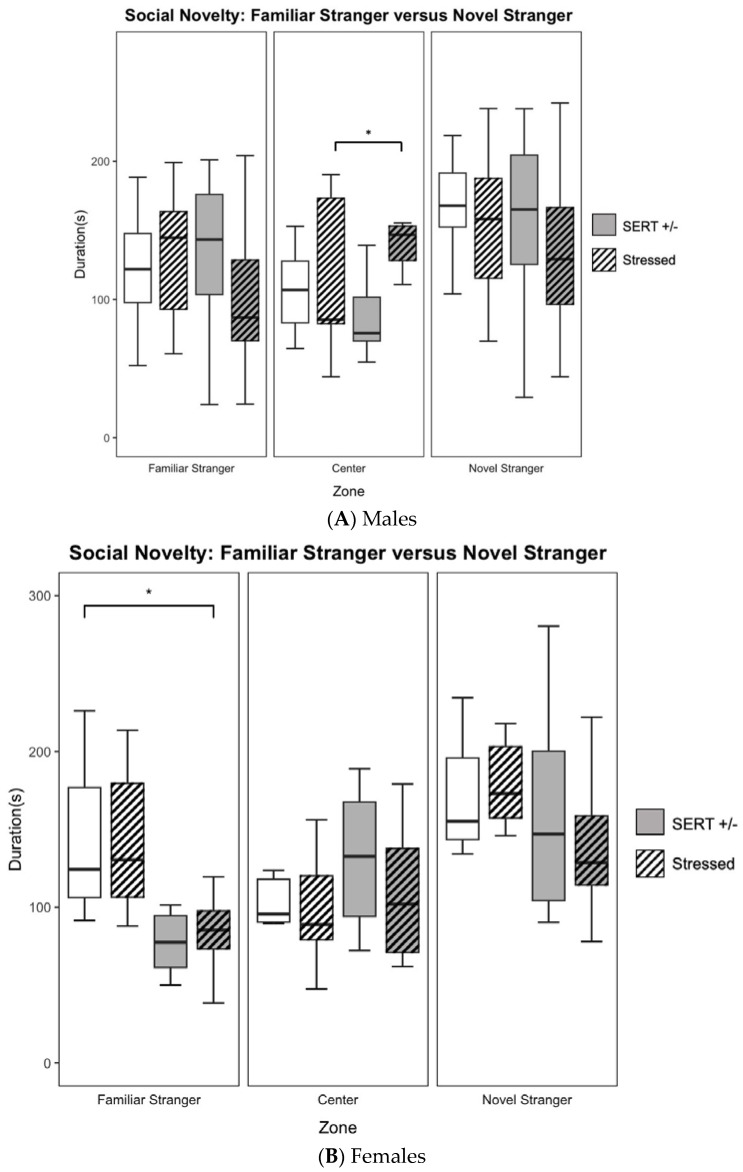
Social behavioral data obtained during the novel stranger versus familiar stranger portion of the social approach task. (**A**). There were no differences found amongst the male groups when comparing the amount of time spent interacting with the novel stranger, the familiar stranger, or time spent in the center chamber for the HS vs WN comparison. (**B**). A priori *t*-testing showed that prenatally stressed female offspring of HS dams spent less time interacting with the familiar stranger than did WN controls. (**C**). Females showed a main effect of genotype (*p* < 0.05), with SERT offspring revealing a higher in SNI than wildtypes. No differences were seen when measuring the proportional ratio of time spent with the novel stranger versus total time spent with either stranger in the male group. (* *p* < 0.05 for individual pairwise comparisons).

**Figure 3 jpm-13-01412-f003:**
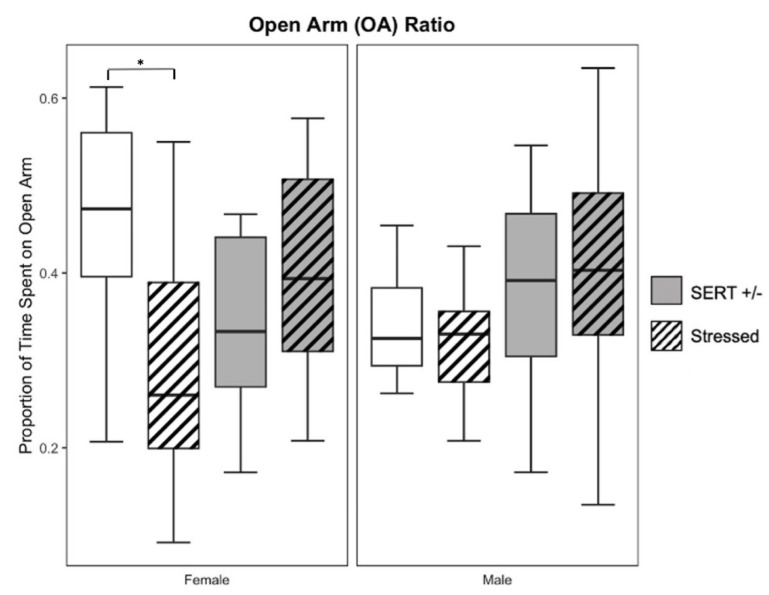
Measures used for anxiety-like behavior include the ratio of time spent in the open arm in comparison to time spent in the closed arm. There was a significant interaction between genotype and stress across the female OA ratios, driven primarily by a difference in stress in the WT offspring. (* *p* < 0.05, for individual pairwise comparisons).

**Figure 4 jpm-13-01412-f004:**
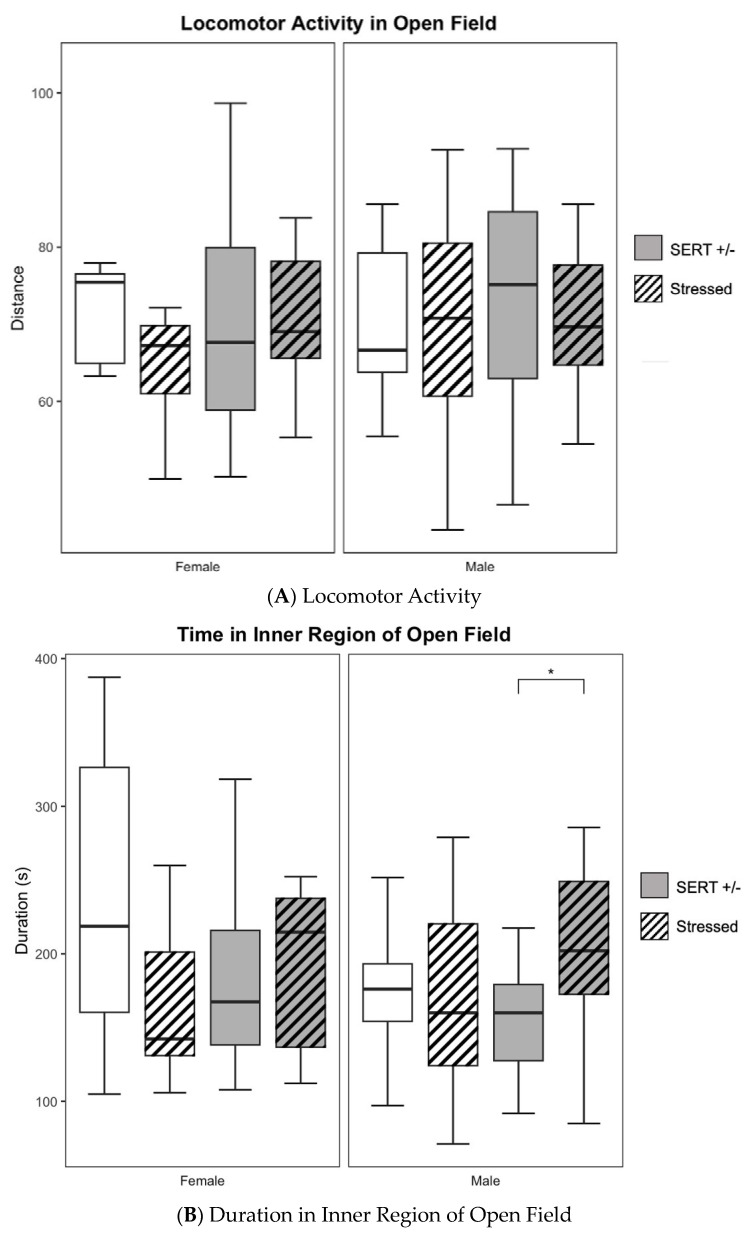
(**A**). No differences were seen across all groups in movement measured by distance traveled during open-field testing. (**B**). HS males also spent significantly more time in the inner region of the open field apparatus than did the HN males. (**C**). No differences were seen in immobility across all conditions and sexes. (* *p* < 0.05, for individual pairwise comparisons).

**Figure 5 jpm-13-01412-f005:**
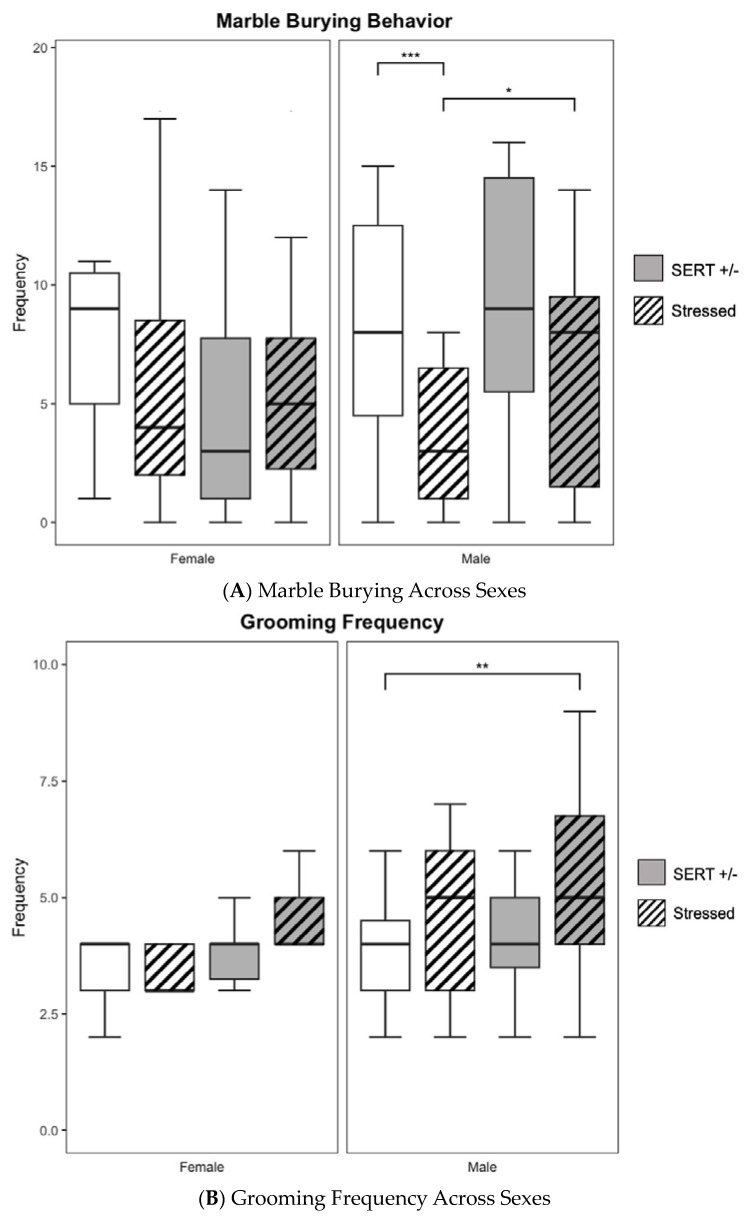
Repetitive behaviors were measured in mice models using frequency of marbles buried, frequency of grooming behavior, and grooming duration. (**A**). Male WS controls buried significantly fewer marbles than did WN and significantly less than did HS mice. (**B**). HS males depicted a significantly higher frequency of self-grooming behavior than WN controls. (**C**). No significant differences were seen across the total grooming duration comparisons. (* *p* < 0.05, ** *p* < 0.01, *** *p* < 0.001, for individual pairwise comparisons).

**Figure 6 jpm-13-01412-f006:**
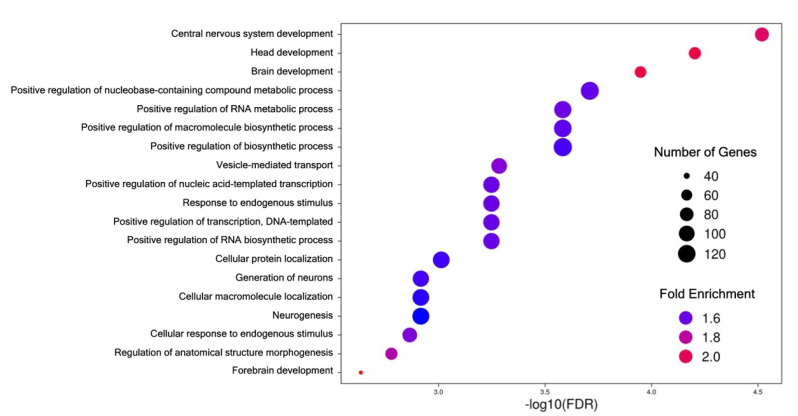
Biological processes GO analysis. The top enriched functional categories based on Gene Ontology (GO) biological processes are depicted based on the 921 unique predicted projections of miRNAs identified at embryonic day 21. Dot size corresponds to the number of identified genes within a given class, and lower enrichment is shown in bluer shades, while increasing enrichment is more red.

**Table 1 jpm-13-01412-t001:** Differentially expressed microRNAs across all conditions.

miRNA	Fold Change (HS vs. WN)	*p*-Value (HS vs. WN)	Fold Change (HS vs. WS)	*p*-Value (HS vs. WS)	Fold Change (HS vs. HN)	*p*-Value (HS vs. HN)	Blood Collection
*mmu-miR-5622-3p*	1.39	0.0075	1.4	0.0219	1.39	0.0054	E21
*mmu-miR-6900-3p*	1.35	0.0333	1.41	0.0286	1.29	0.0436	E21
*mmu-miR-7684-3p*	1.42	0.0015	1.43	0.0022	1.51	0.0037	E21
*mmu-miR-16-5p*	−5.6	0.0487	−12.73	0.0015	−14.53	0.0005	PD60
*mmu-miR-1893*	−3.22	0.0019	−2.96	0.0016	−2.97	0.0015	PD60
*mmu-miR-6347*	−2.52	0.0156	−2.19	0.0405	−2.75	0.0059	PD60
*mmu-miR-126a-3p*	1.23	0.0051	1.23	0.0041	1.25	0.0034	PD60
*mmu-miR-340-5p*	1.31	0.0028	1.44	0.0005	1.28	0.0035	PD60
*mmu-miR-3620-3p*	1.42	0.0045	1.56	0.0014	1.7	0.0002	PD60

List of maternal blood miRNAs identified as differentially expressed on embryonic day 21 (E21) and postnatal day 60 (PD60) in SERT heterozygous—stress (HS) dams relative to three other conditions: wild type no stress (WN), wild type stressed (WS), and SERT heterozygous—no stress (HN).

## Data Availability

Data available upon request.

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
