# Peer review of "microRNA as a Maternal Marker for Prenatal Stress-Associated ASD, Evidence from a Murine Model"

_jpm, 2023, doi:10.3390/jpm13091412_

Round 1

Reviewer 1 Report

The manuscript entitled “microRNA as a Maternal Marker for Prenatal Stress-Associated ASD, Evidence from a Murine Model” by Woo and coworkers is focusing an important topic related to prenatal stress exposure and maternal SERT genotype on off-spring behavior and its association with maternal microRNA (miRNA) expression during pregnancy. The overall quality of this work is good and meets the basic requirement of the journal. Literature searching and citation papers are supporting the main content. Authors concluded that maternal SERT genotype and prenatal stress exposure interact to influence offspring behavior, and that maternal miRNA expression late in pregnancy may serve as a potential marker of a particular subtype of Autism Spectrum Disorder (ASD) pathogenesis. Results showed in the present paper provide important insights into the potential use of miRNAs as a biomarker for ASD risk during pregnancy. 

Major comments:

In my opinion, the Reader has the impression that the transcriptomic results are not sufficiently presented, and lack additional analyzes characterizing all miRNAs detected in the samples. Results from behavioral tests are described in detail with numerous figures and the results from transcriptomics are presented at the end of the results paragraph with one table. The Authors should rewrite this molecular subsection of the results paragraph to balance the accents in the paper. Also, additional tables or biological/molecular GO pathways/KEGG/PANTHER should be introduced to the results. One of the minor remarks is the lack of standardization of the references list.

Author Response

The manuscript entitled “microRNA as a Maternal Marker for Prenatal Stress-Associated ASD, Evidence from a Murine Model” by Woo and coworkers is focusing an important topic related to prenatal stress exposure and maternal SERT genotype on off-spring behavior and its association with maternal microRNA (miRNA) expression during pregnancy. The overall quality of this work is good and meets the basic requirement of the journal. Literature searching and citation papers are supporting the main content. Authors concluded that maternal SERT genotype and prenatal stress exposure interact to influence offspring behavior, and that maternal miRNA expression late in pregnancy may serve as a potential marker of a particular subtype of Autism Spectrum Disorder (ASD) pathogenesis. Results showed in the present paper provide important insights into the potential use of miRNAs as a biomarker for ASD risk during pregnancy. 

Major comments:

In my opinion, the Reader has the impression that the transcriptomic results are not sufficiently presented, and lack additional analyzes characterizing all miRNAs detected in the samples. Results from behavioral tests are described in detail with numerous figures and the results from transcriptomics are presented at the end of the results paragraph with one table. The Authors should rewrite this molecular subsection of the results paragraph to balance the accents in the paper. Also, additional tables or biological/molecular GO pathways/KEGG/PANTHER should be introduced to the results. One of the minor remarks is the lack of standardization of the references list.

As recommended we have added these aspects of the results to the Results section and in Supplementary materials, and standardized the references.  We thank the reviewer for making this a stronger manuscript.

Reviewer 2 Report

The article Woo with co-authors” microRNA as a Maternal Marker for Prenatal Stress-Associated 2 ASD, Evidence from a Murine Model” is devoted to the problem of prenatal stress -associated ASD and the investigation of potential biomarkers like microRNA of ASD pathogenesis.

However, despite the relevance of the ASD problem, the article has a number of major comments.

-Why did the authors not divide by genotypes (into heterozygotes dams and wild type) the offspring from heterozygous females and wild type males into separate group? In my opinion, this this is important in light of the fact that the article was sent to the journal of personalized medicine, and it, in turn, implies just such studies. The authors referred to their work (Jones et al 2010) in which it was written that the emphasis of the study was on the maternal genotype, and not on the offspring genotype. Woo with co-authors didn't mention it in this article.

- The article requires significant revision and corrections. There are a lot of contradictions in the text of the article, especially in the results

 Abstract:

- The sex of the offsprings who found differences in social preference time is not indicated.

In the results showed two figures for this test, and in the abstract it is not clear what specific parameter the authors mean.

In abstract written that (line 28-29)   SERT-het/stress offspring showed significantly longer 28 grooming duration compared to wildtype control offspring, suggesting elevated repetitive behavior. (no sex indication).

But in results fig 5B is shown that male of HS had more grooming frequency compared with WN and fig 5C is shown that mice of both sex has no significant differences in grooming duration and authors wrote about (line 411-413 C. No significant differences were seen across the total grooming duration comparisons).

The authors in the abstract contradict themselves

Introduction

Introduction written quite well. The authors referred to their previous work where the same animal model was used. But as I wrote above, the authors should indicate that, as in the previous one, the emphasis was on the maternal genotype and the genotype of the offspring was not taken into account (although this is a big lack of this work).

Materials and methods

Part –Animals

why did the authors introduced this abbreviation if they don’t used it anywhere else –line 126 - SERT-heterozygous (het).

In section prenatal chronic variable stress authors should introduced abbreviation of wildtype no stress, wildtype stress, SERT-het no stress, and SERT-het stress groups that

wildtype no stress (WN),

wildtype stress (WS),

SERT-het no stress (HN),

and SERT-het stress (HS)

Since in the text some of the abbreviations have explanations, and some appear without decoding, and add all these abbreviations to the figures and enter the designations of all groups in the figures.

The statistics section needs to be expanded and described in more detail. There is no description of how and why some values were removed from the tests.

-maternal genotype x stress exposure and genotype x stress it is the same factors? Line 243-244

Results

The authors obtained 83 males and 42 females which is unusual. The authors did not comment this fact. Is it possible that stress can influence sex distribution?

It is also needed to add how many animals were received in each group

The authors need to rewrite the social approach section Novel stranger versus familiar stranger

The authors wrote that – (line 307-309) While males did not show any significant main effects in SNI, females revealed a main effect of genotype (F 1,38 = 4.45, p < 0.05, η2 = 0.10, see Fig 2C) in SNI scores. But in figure 2C in the figure caption they wrote – (Line 338-340) -C. No differences were seen when measuring the proportional ratio of time spent with the novel stranger versus total time spent with either stranger across both male and female groups.

And line 311-314 they wrote that - However, HS males (M= 147.42, S.E.M = 14.47) spent significantly (t48.692= 2.139, p=0.038, unpaired t-test) less time than their WN counterparts ( M = 184.64, S.E.M = 11.72) engaging with the novel wildtype mouse, but this pattern was not seen in our testing of the female offspring, see Figures 2A-B. But on figure 2A-B

 On Figure is shown the exact opposite.

Elevated Plus Maze

Line 342 -Open arm (OA) ratio was used to quantify anxiety – like behavior in male (n = 67)… Where another 13 males?

This part needs also rewriting and in fig 3 also needs add time in the OA (because female had significant differences) and in closes arms also.

Again, the contradiction between the description of the results in the text and what is shown in the figure 3

Line-345-346- Females on the other hand did show a significant interaction effect between genotype and stress (F1,38 = 7.41, p < 0.01, η2 = 0.16, see Figure 3) in their respective OA ratios.

And line 358 legend of figure 3 - A. There was a significant interaction across the female OA ratios

And figure 3. No differences are shown

Discussion

The discussion section needs to be completely rewritten, firstly, because there is a lot of confusion in the results and very little discussion of the results obtained from animal behavior. The results obtained in the marble burying test are not discussed in any way, although the result was obtained contrary to what the authors assumed - stress reduces the number of marbles buried. in ASD- more marbles should be buried (Amodeo et al 2012, Yang et al., 2007; Yang, Zhodzishsky, Crawley, 2007; McFarlane et al., 2008, Jacome и др., 2011; Burket et al., 2015b; Burket et al., 2015).

There is no discussion of the differences obtained by sex at all. Much has been given to the potential role of microRNA, although the authors did not find any correlations with behavior.

Conclusion. -The article is very sloppy. The article Woo with co-authors has good enough publishing potential, but needs a lot of revisions across all sections. Authors should carefully check all their data and their presentation.

.

Author Response

The article Woo with co-authors” microRNA as a Maternal Marker for Prenatal Stress-Associated 2 ASD, Evidence from a Murine Model” is devoted to the problem of prenatal stress -associated ASD and the investigation of potential biomarkers like microRNA of ASD pathogenesis.

However, despite the relevance of the ASD problem, the article has a number of major comments.

-Why did the authors not divide by genotypes (into heterozygotes dams and wild type) the offspring from heterozygous females and wild type males into separate group? In my opinion, this this is important in light of the fact that the article was sent to the journal of personalized medicine, and it, in turn, implies just such studies. The authors referred to their work (Jones et al 2010) in which it was written that the emphasis of the study was on the maternal genotype, and not on the offspring genotype. Woo with co-authors didn't mention it in this article.

- The article requires significant revision and corrections. There are a lot of contradictions in the text of the article, especially in the results

We have added the comment that the emphasis is on maternal genotype at the end of the Introduction and added this in the limitations.  It is based on our previous work, and the lack of statistical power to allow us to add yet another statistical group.  We have addressed the contradictions as described below.

 Abstract:

- The sex of the offsprings who found differences in social preference time is not indicated.

In the results showed two figures for this test, and in the abstract it is not clear what specific parameter the authors mean.

The sex was listed, but we have edited to make it more clear, and have specified which specific parameters we are discussin g in the Abstact.

In abstract written that (line 28-29)   SERT-het/stress offspring showed significantly longer 28 grooming duration compared to wildtype control offspring, suggesting elevated repetitive behavior. (no sex indication).

But in results fig 5B is shown that male of HS had more grooming frequency compared with WN and fig 5C is shown that mice of both sex has no significant differences in grooming duration and authors wrote about (line 411-413 C. No significant differences were seen across the total grooming duration comparisons).

The authors in the abstract contradict themselves

We have also clarified the specific measure in the Abstract and made the sex more clear for grooming and clarified the grooming measure findings- it was only frequency that was different. Part of the problem is that significance was not consistently marked in the figures, which is now addressed.

Introduction

Introduction written quite well. The authors referred to their previous work where the same animal model was used. But as I wrote above, the authors should indicate that, as in the previous one, the emphasis was on the maternal genotype and the genotype of the offspring was not taken into account (although this is a big lack of this work).

As above, we have added the emphasis on maternal genotype at the end of the Introduction and in the limitations.  It is based on our previous work, and the lack of statistical power to allow us to add yet another statistical group.

Materials and methods

Part –Animals

why did the authors introduced this abbreviation if they don’t used it anywhere else –line 126 - SERT-heterozygous (het).

We actually do use this abbreviation in the subsequent paragraph to help streamline the group descriptions in the next paragraph.

In section prenatal chronic variable stress authors should introduced abbreviation of wildtype no stress, wildtype stress, SERT-het no stress, and SERT-het stress groups that

wildtype no stress (WN),

wildtype stress (WS),

SERT-het no stress (HN),

and SERT-het stress (HS)

Since in the text some of the abbreviations have explanations, and some appear without decoding, and add all these abbreviations to the figures and enter the designations of all groups in the figures.

 We have added these abbreviations to the figure legends as suggested.

The statistics section needs to be expanded and described in more detail. There is no description of how and why some values were removed from the tests.

Sorry for the lack of clarity- there were different numbers between tasks due to missing data.  No values were removed.

-maternal genotype x stress exposure and genotype x stress it is the same factors? Line 243-244

Yes, this is just a more specific way of stating the gene x stress interaction.

Results

The authors obtained 83 males and 42 females which is unusual. The authors did not comment this fact. Is it possible that stress can influence sex distribution?

We have added discussion of this odd sex distribution.

It is also needed to add how many animals were received in each group

This is now specified.

The authors need to rewrite the social approach section Novel stranger versus familiar stranger

Good point- this is fixed.

The authors wrote that – (line 307-309) While males did not show any significant main effects in SNI, females revealed a main effect of genotype (F 1,38 = 4.45, < 0.05, η2 = 0.10, see Fig 2C) in SNI scores. But in figure 2C in the figure caption they wrote – (Line 338-340) -C. No differences were seen when measuring the proportional ratio of time spent with the novel stranger versus total time spent with either stranger across both male and female groups.

And line 311-314 they wrote that - However, HS males (M= 147.42, S.E.M = 14.47) spent significantly (t48.692= 2.139, p=0.038, unpaired t-test) less time than their WN counterparts ( = 184.64, S.E.M = 11.72) engaging with the novel wildtype mouse, but this pattern was not seen in our testing of the female offspring, see Figures 2A-B. But on figure 2A-B

 On Figure is shown the exact opposite.

We apologize for the clarity issues- we have added text to more completely describe the Figure that we hope resolves this confusion and a major problem is that several figures were uploaded that failed to mark the significant results, which is now fixed.

Elevated Plus Maze

Line 342 -Open arm (OA) ratio was used to quantify anxiety – like behavior in male (n = 67)… Where another 13 males?

This part needs also rewriting and in fig 3 also needs add time in the OA (because female had significant differences) and in closes arms also.

Again, the contradiction between the description of the results in the text and what is shown in the figure 3

Line-345-346- Females on the other hand did show a significant interaction effect between genotype and stress (F1,38 = 7.41, < 0.01, η2 = 0.16, see Figure 3) in their respective OA ratios.

And line 358 legend of figure 3 - A. There was a significant interaction across the female OA ratios

And figure 3. No differences are shown

Again, figures were uploaded that did not include the marking of the significant results.  This has been fixed.  Some of the main effects and interactions are not on the Figure, but the individual differences on t-tests are now marked.  As is now indicated in the Results, not all of the mice could complete this task, so the sample was lower.

Discussion

The discussion section needs to be completely rewritten, firstly, because there is a lot of confusion in the results and very little discussion of the results obtained from animal behavior. The results obtained in the marble burying test are not discussed in any way, although the result was obtained contrary to what the authors assumed - stress reduces the number of marbles buried. in ASD- more marbles should be buried (Amodeo et al 2012, Yang et al., 2007; Yang, Zhodzishsky, Crawley, 2007; McFarlane et al., 2008, Jacome Ð¸ Ð´Ñ€., 2011; Burket et al., 2015b; Burket et al., 2015).

There is no discussion of the differences obtained by sex at all. Much has been given to the potential role of microRNA, although the authors did not find any correlations with behavior.

The clarification of the Results and fixing the Figures we hope has resolved the confusion in the Discussion.  Additionally, we have discussed the unexpected marble burying finding, and highlighted the sex differences, but we do acknowledge that there is no correlation with behavior- but sample size may be an issue. 

Conclusion. -The article is very sloppy. The article Woo with co-authors has good enough publishing potential, but needs a lot of revisions across all sections. Authors should carefully check all their data and their presentation.

We hope this addressed the concerns.

Round 2

Reviewer 2 Report

In general, the article of Woo with co-authors” microRNA as a Maternal Marker for Prenatal Stress-Associated 2 ASD, Evidence from a Murine Model” has acquired a decent seed for publication and can be published after correcting the comments

comments.

Line 24 why the abbreviation SER-het/stress was removed? Its appear on line 27

Line 92 – no abbreviation chronic variable stress. In abstract were added abbreviation CVS. In text also will need add this.

It is need to indicate in the statistics section how the data is presented in the figures. You can see from them that you have quartiles and medians marked there. In the text you present the data as MEAN±SEM

Line 301 enter the abbreviation SPI instead Social preference index

Line 454 remove A

Line 454 on figure 4 time mobile is not presented

Line 539 What behavioral testing were in dams?  It is not a very clear phrase

Line 569 abbreviation miRNAs were added in line 65

Discussion

In the discussion, there was little discussion of the results obtained during testing of the offspring. The authors described the results very modestly. There is no discussion of the differences obtained by sex at all

Authors should also carefully review the abbreviations they introduce throughout the text. Sometimes they do it again.

Author Response

In general, the article of Woo with co-authors” microRNA as a Maternal Marker for Prenatal Stress-Associated 2 ASD, Evidence from a Murine Model” has acquired a decent seed for publication and can be published after correcting the comments

comments.

Line 24 why the abbreviation SER-het/stress was removed? Its appear on line 27

We replaced SERT-het back in line 24 as the abbreviation for the genotype of SERT heterozygous knockouts. The later use of SERT-het/stress is the first time combining the genotype and exposure to stress variables in the abstract.

Line 92 – no abbreviation chronic variable stress. In abstract were added abbreviation CVS. In text also will need add this.

We added the abbreviation to line 92.

It is need to indicate in the statistics section how the data is presented in the figures. You can see from them that you have quartiles and medians marked there. In the text you present the data as MEAN±SEM

We have clarified that our written statistics depict MEAN±SEM while the figures use boxplots which highlights different measurements.

Line 301 enter the abbreviation SPI instead Social preference index

These are fixed.

Line 454 remove A

Line 454 on figure 4 time mobile is not presented

Line 539 What behavioral testing were in dams?  It is not a very clear phrase

Line 569 abbreviation miRNAs were added in line 65

These are removed and corrected.

Discussion

In the discussion, there was little discussion of the results obtained during testing of the offspring. The authors described the results very modestly. There is no discussion of the differences obtained by sex at all

We have added more information reviewing the parallels between the behaviors of male and female offspring.

Authors should also carefully review the abbreviations they introduce throughout the text. Sometimes they do it again.

We thank you for bringing this to our attention, and we have thoroughly reviewed the manuscript for inconsistent abbreviations based on the MDPI abbreviations rules throughout the abstract, main body, and figures/tables.